# Celiac Immunogenic Potential of α-Gliadin Epitope Variants from *Triticum* and *Aegilops* Species

**DOI:** 10.3390/nu11020220

**Published:** 2019-01-22

**Authors:** Ángela Ruiz-Carnicer, Isabel Comino, Verónica Segura, Carmen V. Ozuna, María de Lourdes Moreno, Miguel Ángel López-Casado, María Isabel Torres, Francisco Barro, Carolina Sousa

**Affiliations:** 1Departamento de Microbiología y Parasitología, Facultad de Farmacia, Universidad de Sevilla, 41012 Sevilla, Spain; acarnicer@us.es (Á.R.-C.); icomino@us.es (I.C.); vsegura@us.es (V.S.); lmoreno@us.es (M.d.L.M.); 2Departamento de Mejora Genética Vegetal, Instituto de Agricultura Sostenible (IAS-CSIC), 14004 Córdoba, Spain; carmen.ozuna@eez.csic.es (C.V.O.); fbarro@ias.csic.es (F.B.); 3Departamento de Gastronterología pediátrica, Hospital Virgen de las Nieves, 18014 Granada, Spain; drlopezcasado@digestivointegral.es; 4Departamento de Biología Experimental, Campus Universitario Las Lagunillas, 23071 Jaén, Spain; mitorres@ujaen.es

**Keywords:** celiac disease, α-gliadin, 33-mer, DQ2.5-glia-α1, DQ2.5-glia-α2, DQ2.5-glia-α3 epitopes, wheat species

## Abstract

The high global demand of wheat and its subsequent consumption arise from the physicochemical properties of bread dough and its contribution to the protein intake in the human diet. Gluten is the main structural complex of wheat proteins and subjects affected by celiac disease (CD) cannot tolerate gluten protein. Within gluten proteins, α-gliadins constitute the most immunogenic fraction since they contain the main T-cell stimulating epitopes (DQ2.5-glia-α1, DQ2.5-glia-α2, and DQ2.5-glia-α3). In this work, the celiac immunotoxic potential of α-gliadins was studied within Triticeae: diploid, tetraploid, and hexaploid species. The abundance and immunostimulatory capacity of CD canonical epitopes and variants (with one or two mismatches) in all α-gliadin sequences were determined. The results showed that the canonical epitopes DQ2.5-glia-α1 and DQ2.5-glia-α3 were more frequent than DQ2.5-glia-α2. A higher abundance of canonical DQ2.5-glia-α1 epitope was found to be associated with genomes of the BBAADD, AA, and DD types; however, the abundance of DQ2.5-glia-α3 epitope variants was very high in BBAADD and BBAA wheat despite their low abundance in the canonical epitope. The most abundant substitution was that of proline to serine, which was disposed mainly on the three canonical DQ2.5 domains on position 8. Interestingly, our results demonstrated that the natural introduction of Q to H at any position eliminates the toxicity of the three T-cell epitopes in the α-gliadins. The results provided a rational approach for the introduction of natural amino acid substitutions to eliminate the toxicity of three T-cell epitopes, while maintaining the technological properties of commercial wheats.

## 1. Introduction

Wheat is one of the most widely cultivated cereals in the world and constitutes a major source of energy, protein, and fiber in the diet. Increasing global demand for wheat and its subsequent consumption, with an annual production of about 750 million tons, is due to its unique viscoelastic properties for its inclusion in food products and to industrialization and westernization [1,2,3]. The wheat group has evolved through allopolyploidization, that is, through hybridization between species from the genera *Aegilops* and *Triticum* followed by genome doubling [4]. Genetic studies have provided valuable information regarding which wild cereal species are the relatives of modern domesticated cereals, and which geographical wild plant produced the domesticated forms that are used in food production today [5]. The diploid wild wheat that was first domesticated is thought to have been *Triticum monococcum (A^m^A^m^*), which is still growing in some parts of the world both for animal feed and human consumption. Wheats with more than one genome are known as polyploid wheats. The AA genome of the tetraploid wheats is closely similar to that of *T. urartu*, and the BB genome is related to *Aegilops speltoides* (BB). The wild tetraploid, formed after the hybridization, was designated as *Triticum turgidum* ssp. *dicoccoides* (wild emmer; BBAA), and the first domesticated tetraploid was *T. turgidum* ssp. *dicoccum* (cultivated emmer; BBAA), from which, the cultivated *T. turgidum* ssp. *durum* has evolved. The hexaploid wheat, *Triticum aestivum* ssp. *aestivum* (BBAADD), consists of three genomes designated A, B, and D. The A and B genomes of hexaploid wheats come from the A and B genomes of tetraploid wheat. The hexaploid wheats resulted from the hybridization of cultivated emmer and a wild grass species identified as *Aegilops tauschii* (DD), followed by polyploid formation which gave rise to a new species that has three genomes designated BBAADD [6,7]. The main wheat species grown throughout the world is the hexaploid *T. aestivum*, usually called “common” or “bread” wheat. In terms of total production, the next variety in importance is the tetraploid durum or macaroni wheat (*T. turgidum* L. subsp. *durum* Desf.). This is adapted to hot dry climates and is widely used for the production of pasta. Common wheat species account for nearly 94% of the total production, with durum wheat representing 5%, and other wheat forms about 1% [6,8]. 

Although wheat has always been recognized as a fundamental food, this cereal cannot be tolerated by certain individuals since it is responsible for significant pathologies, called gluten-related disorders, such as celiac disease (CD), wheat allergy, non-celiac gluten sensitivity, gluten ataxia, and dermatitis herpetiformis [9]. CD is an immune-mediated systemic disorder elicited by the ingestion of gluten in genetically susceptible individuals. It affects around 1% of the global population and is based on a variable combination of intestinal and extra-intestinal signs and symptoms, celiac specific antibodies, HLA-DQ2/8 haplotypes, and enteropathy [9,10,11]. Gluten proteins are rich in proline and glutamine residues, which make them resistant to being fully digested in the gastrointestinal track. Partial digestion of gluten generates small peptides that provoke autoimmune disorders in celiac people. The most accepted model for explaining CD immunopathogenesis is the two-signal model [12] characterized by a first innate immune response followed by a secondary antigen-specific adaptive response. According to this model, certain peptides, such as the 19-mer gliadin peptide, trigger an innate immune response [13] mainly characterized by the production of interleukin 15 (IL-15) by epithelial cells. The result is the disruption of the epithelial barrier by increasing the permeability and inducing enterocyte apoptosis [14]. As a consequence, the immune-adaptive peptides, like the 33-mer, can now reach the lamina propria where they are deaminated by the tissue transglutaminase (tTG2). Such deamidation provides a negative charge to gliadin peptides and hence enhances their affinity to bind within the HLA-DQ2/8 bound, which is also the ‘susceptibility gene’ in CD, expressed on the surface of dendritic cells (DCs) [15,16,17]. DCs are therefore central in CD pathogenesis since they present a gluten antigen to T cells, [18] thereby driving progression of the pro-inflammatory antigen-specific adaptive immune response, which will turn into the symptomatology of the disease.

Gluten is a complex mixture of storage proteins of cereals such as wheat, rye, barley, oats, and their hybrid derivatives. Gluten proteins have been classified according to their solubility [19]. In wheat, these proteins are defined as gliadins (soluble in 60–70% ethanol) and glutenins (only soluble under stronger conditions, i.e. acids, reducing agents and detergents, urea, etc.) [20]. According to their electrophoretic mobilities, gliadins are divided into three groups: α- and β-gliadins, γ-gliadins, and ω-gliadins [19], while the glutenins are divided into the high molecular weight (HMW) and the low molecular weight (LMW) glutenin subunits (GSs) [21,22]. Among the gliadins, the α-gliadins have the strongest immunogenicity [23], and four T-cell stimulatory epitopes have been identified as being responsible for eliciting the immunogenicity of α-gliadin. Two of these are the major epitopes and they are present in the 33-mer peptide, which is the main contributor to the immunogenicity of the gluten [24] and contains six copies of these two overlapping T-cell epitopes: three copies of the DQ2.5-glia-α1 and three copies of the DQ2.5-glia-α2. The other two T-cell stimulatory epitopes are minor epitopes: DQ2.5-glia-α3 and DQ8-glia-α1 [24,25,26,27]. However, natural substitutions of these canonical epitopes could also contribute to the toxicity of wheat [28], and it could suggest that the total CD immunogenicity of gluten protein is a result of the canonical epitopes and their variants, some of which are more abundant than the canonical epitopes themselves. Gluten can have different immunogenic potential sequences whose proportions in each species are also variable. For this reason, it is important to study the amino acid substitutions in the variants of these epitopes; interestingly, these variants could increase, reduce, or suppress the CD response.

In earlier work, next-generation sequencing and Sanger sequencing of α-gliadins from diploid and polyploid wheats provided six types of α-gliadins with major differences in their frequencies. The canonical CD epitopes and their variants were identified in the different types of α-gliadins [29]. In the present study, we used the sequence data with one or two mismatches and canonical epitopes obtained in Ozuna et al. [29], and we have built upon the previous research by exploring the abundance of different DQ2.5-glia-α1, DQ2.5-glia-α2, and DQ2.5-glia-α3 epitope variants per species in diploid and polyploid wheats. Moreover, the immunogenic potential of these epitope variants in wheat species was studied by testing their binding capacity to anti-33-mer monoclonal antibodies (moAbs) [30,31] and to induce T-cell proliferation. The anti-33-mer antibodies were able to detect the presence of gliadin 33-mer related epitopes in prolamins from wheat, barley, rye, and various oats varieties as well as in food samples and human samples to monitor gluten free diet (GFD) compliance and transgressions [32]. Our study showed that the canonical epitopes DQ2.5-glia-α1 and DQ2.5-glia-α3 were more frequent than DQ2.5-glia-α2. The most abundant natural modification was found in the DQ2.5-glia-α3 domain in all the sequences studied. However, this variant decreased its immunogenicity with respect to the canonical epitope. On the other hand, one of the most representative variants of DQ2.5-glia-α2 (40%) showed an immunogenicity equivalent to the canonical epitope. Our results provide a rational approach for the introduction or selection of natural amino acid substitutions to eliminate the toxicity of three α-gliadin T-cell epitopes, while keeping the technological properties of the commercial wheats.

## 2. Materials and Methods 

### 2.1. Catalogue and Abundance of CD Epitopes from Diploid, Tetraploid, and Hexaploid Wheat Varieties

Canonical epitopes DQ2.5-glia-α1 (PF/YPQPQLPY), DQ2.5-glia-α2 (PQPQLPYPQ), and DQ2.5-glia-α3 (FRPQQPYPQ) and variants with one or two mismatches provided by Ozuna et al. [29] were obtained from diploid, tetraploid, and hexaploid wheats (Figure 1).

The frequency/abundance of each peptide in the sequences of the different wheats was studied *in silico*. The abundance of each epitope was calculated by multiplying the total number of epitopes found in a given gene by the frequency of that gene in the genome.

These canonical CD epitopes and their most representative variants with one or two mismatches were synthesized as deaminated and non-deaminated 9-mer peptides. The peptides were supplied by Biomedal S.L. (Seville, Spain).

### 2.2. Enzyme-Linked Immunosorbent Assay (ELISA)

Maxisorp microtitre plates (Nunc, Roskilde, Denmark) were coated with gliadin solution (Sigma, St Louis, MO, USA) and incubated overnight at 4 °C. The plates were washed with phosphate-buffered saline (PBS) containing 0.05% Tween 20 and blocked with PBS-bovine serum albumin (BSA) 3% for 1 h at room temperature (RT), and 33-mer peptide was used as standard. Serial dilutions of peptides were made, to each of which horseradish peroxidase (HRP)-conjugate with anti-33-mer antibody (moAb) was added [32]. The samples were pre-incubated at RT for 1 h and then added to the wells. After 1 h of incubation at RT, the plates were washed and substrate solution (TMB, Sigma) was added. The reaction was stopped at 15 min with 1 M sulfuric acid and the absorbance at 450 nm was measured (microplate reader UVM340; Asys Hitech GmbH, Eugendorf, Austria). Two separate assays were performed, each with two repetitions.

### 2.3. Peripheral Blood Mononuclear Cells (PBMCs) and Cell Cultures

Peripheral blood mononuclear cells from 18 child patients with active CD on a gluten-containing diet were isolated from 6 mL of heparinized blood by Histopaque gradient centrifugation and cultured at a density of 1 × 10^6^ cells per milliliter in 96-multiwell culture plates in RPMI-1640 culture medium (Sigma-Aldrich) supplemented with 10% fetal bovine serum (GIBCO-Invitrogen Ltd), 1% penicillin-streptomycin, and 0.1% gentamicin (Sigma-Aldrich). After 48 h, PBMCs were incubated with different peptides (50 µg/mL). After 48 h of stimulation, the free supernatants were collected and stored at −80 °C until the interferon gamma (IFN-γ) analyses were carried out. 

### 2.4. Cell Proliferation Analysis

T-cell proliferation was determined after 48 h of incubation using the ELISA 5-bromo-2-deoxyuridine (BrdU) cell proliferation test (Millipore Chemicon, Temecula, California, USA). The stimulation index (SI) value was calculated by dividing the mean absorbance at 450 nm after stimulation by the mean absorbance of T cells exposed to the culture medium alone (negative control). 

### 2.5. IFN-γ Production

Supernatants from the PBMC culture were collected after 48 h and stored at 80 °C for IFN- γ determination using a commercial ELISA kit in accordance with the manufacturer’s instructions (Thermo Scientific, Madrid, Spain). Standards were run on each plate. The sensitivity of the assay was <2 pg/mL.

### 2.6. Statistical Analysis of T Cells and IFN-γ Assays

Each experiment was carried out in duplicate on separate days. The data is expressed as mean and *SD*. All statistical analyses were performed with the STATGRAPHICS Centurion XVI program. The analysis of variance (ANOVA), followed by the Tukey test for mean multiple comparison, was used. In this study, *p* values lower than 0.05 (*p* < 0.05) were considered significant.

## 3. Results and Discussion

### 3.1. Relative Abundance of DQ2.5-Glia-α1, DQ2.5-Glia-α2, and DQ2.5-Glia-α3 Domains and Their Variants in Triticum and Aegilops Species

The complete repertoire of peptides involved in the pathogenesis of CD remains a daunting task due to the great heterogeneity of gluten proteins [23,26]. Several studies have demonstrated that peptides derived from α-gliadins induce the strongest T-cell responses in the vast majority of patients [23,33,34,35]. The α-gliadins can have different sequences and their proportions in each species are also variable. In the present study, we have explored the abundance of different DQ2.5-glia-α1, DQ2.5-glia-α2, and DQ2.5-glia-α3 variants in 96 genotypes from diploid and polyploid wheats. Among these genotypes, 27 accessions were commercial lines and 69 were non-commercial lines (Figure 1).

The DQ2.5-glia-α epitopes are located in the 33-mer region of α-gliadins (Figure 2a). Although seventy-eight variants were found for these three canonical epitopes across the Triticeae species [29], only the most representative variants (covered by >80 reads), encompassing one or two mismatches, were used for this study; of which 9 variants were from DQ2.5-glia-α1, 10 from DQ2.5-glia-α2, and 14 from DQ2.5-glia-α3 (Figure 2b). 

In view of the total abundance of the different canonical epitopes, DQ2.5-glia-α1 and DQ2.5-glia-α3 were more abundant than DQ2.5-glia-α2 (*p* < 0.05, Figure 3). There were no significant differences between the abundance of DQ2.5-glia-α1 and DQ2.5-glia-α3, however, we found higher variability of the DQ2.5-glia-α1 canonical epitope in hexaploid wheats, since its abundance fluctuated widely depending on the different hexaploid species, while it remained evenly distributed in the DQ2.5-glia-α3 canonical epitope (*p* = 0.02, Figure 3). 

Figure 4 shows the abundance of CD canonical epitopes and variants per species. The percentage of DQ2.5-glia-α1 canonical epitope with respect to variants was 80%. This epitope was present in all wheat genomes with the exception of BB diploids. The highest abundance was found in *T. compactum*, *T. monococcum*, and *Ae. tauschii*. The most abundant variant (range from 0.1% to 20%) was P_1_Y_2_P_3_Q_4_P_5_Q_6_L_7_**F_8_P**_9_ with two mismatches (P to F at p8 and Y to P at p9). This variant was present in all wheat genomes with the exception of BB and DD diploids. The next most abundant variant was the substitution of P to L at p5, but this variant was present in only BBAA and BB genomes (Figure 5).

The percentage of DQ2.5-glia-α2 canonical epitope with respect to the different variants of this epitope was 14%, and this epitope was only present in hexaploids BBAADD and DD diploids. This finding may indicate that this epitope came from *Ae*. *tauschii*, the donor of the D genome to bread wheat. The DQ2.5-glia-α2 variants P_1_Q_2_P_3_Q_4_L_5_P_6_Y_7_**S_8_**Q_9_ and P_1_Q_2_P_3_Q_4_**P_5_Q_6_**Y_7_P_8_Q_9_ were the most frequent (80%). The highest abundance score (range from 46% to 74%) of P_1_Q_2_P_3_Q_4_L_5_P_6_Y_7_**S_8_**Q_9_ (P to S substitution at p8) occurred in AA diploids and was absent from DD and BB diploid genomes. In contrast, P_1_Q_2_P_3_Q_4_**P_5_Q_6_**Y_7_P_8_Q_9_ (L to P at p5 and P to Q at p6), with two mismatches, presented high abundance in all genomes, with the exception of *T. monococcum* (A^m^A^m^ diploid) (Figure 4 and Figure 5). 

Regarding the DQ2.5-glia-α3, the epitope variant F_1_**P_2_**P_3_Q_4_Q_5_P_6_Y_7_P_8_Q_9_ (with R to P substitution at p2) was the most frequent, with an abundance greater than 75% across all species, except in *Triticum polonicum* (BBAA) and *Triticum urartu* (A^u^A^u^), with abundances of 67.9% and 42.2%, respectively. The second and third most frequent variants, F_1_**L_2_**P_3_Q_4_**L_5_**P_6_Y_7_P_8_Q_9_ (R to L at p2 and Q to L at p5) and F_1_**P_2_**P_3_Q_4_Q_5_**S_6_**Y_7_P_8_Q_9_ (R to P at p2 and P to S at p6), had two mismatches and were absent in AA and DD diploid genomes, which could indicate that the BB genome is the origin of this variant in the polyploid varieties; in fact, the abundance of this epitope variant in the remaining genomes was very similar (≈20%) (Figure 4 and Figure 5). 

The process of hybridization between *Ae. tauschii* and *T. dicoccum* provided the DD genome, and new gluten gene combinations, to hexaploid wheats, thereby considerably improving their bread baking properties compared to that of tetraploid wheats, particularly the HMW-glutenin subunits [36]. However, the DD genome also encodes for gliadins that have been reported as highly immunogenic, as the DD genome has the highest number of potential immunogenic α-gliadin peptides [37], while those from the BB genome contribute the least [38,39]. We found that the three canonical epitopes are present in the DD genome, with a representation ranging from 43% to 65%. In hexaploids (BBAADD), all canonical epitopes are also present, but in a smaller proportion (<40%) than the DD genome. In contrast, in the AA genome, only DQ2.5-glia-α1 and DQ2.5-glia-α3 are present, and the BB genome is not represented by any of the canonical epitopes. 

### 3.2. Anti-33-mer MoAb Binding Capacity and T-cell Stimulatory of DQ2.5-α-Gliadin-Derived Peptides 

Several of the amino acid variants that we found in the α-gliadin epitope sequences had never been described previously, while a number had been described but had never been tested for their immunogenic and stimulatory capacities. In order to determine which variants are capable of inducing a CD stimulatory response, the variants from DQ2.5-glia-α epitopes were synthesized as native and deaminated peptides and tested for their capacity to bind to anti-33-mer monoclonal antibodies (moAbs) and to induce T-cell proliferation, respectively (Figure 6). The latter was confirmed with gamma interferon assays (IFN-γ). The positioning of deamidated glutamine residues is strongly related to the positioning of proline residues, which is particularly strict in the case of DQ2.5 epitopes (but not DQ8 epitopes), as DQ2.5 only accepts proline at a certain position in the peptide binding groove [26,40]. The capacity of DQ2.5-glia-α epitopes to trigger proliferation of T cells was tested in deaminated peptides, deamidation of glutamine (Q) at p6 in DQ2.5-glia-α1 domain, p4 in DQ2.5-glia-α2 domain, and p4 in DQ2.5-glia-α3 domain.

The DQ2.5-glia-α1 and DQ2.5-glia-α2 epitopes were regarded as major CD epitopes, as they are recognized by most of CD patients [41]. The anti-33-mer moAbs reacted strongly with the canonical DQ2.5-glia-α1 epitope P_1_{F/Y}_2_P_3_Q_4_P_5_Q_6_L_7_P_8_Y_9_. In comparison with the canonical CD epitope, the variants P_1_F_2_**L_3_**Q_4_P_5_Q_6_L_7_P_8_Y_9_ (proline (P) to leucine (L) substitution at p3) and P_1_F_2_**S_3_**Q_4_P_5_Q_6_L_7_P_8_Y_9_ (P to serine (S) substitution at p3) showed a cross-reactivity (CR) of 37.7% and 55.4%, respectively. With regard to studies of PBMCs in DQ2.5-glia-α1 variants, the single substitution P to S at p3 maintained similar stimulation capacity to that of the canonical epitope, although it was not abundant (0.2% to 0.8%) in wheat species. Similarly, the variant with P to L substitution at p3 was very low in abundance (0.1% to 1.6%) and was found in all the polyploid species except for *T. spelta,* while in diploids it was only found in species with the AA genome. However, substitutions of Q to histidine (H) at p4 and p6, respectively, abolished the stimulatory capacity of this epitope, probably because it provides a positive charge or via its influence in the deamidation at p6, as previously observed by Schumann et al. [42]. Moreover, P to L at p8, or two substitutions, also abolished the stimulatory capacity and showed no affinity for the moAbs (Figure 6a). 

As indicated in Figure 6b, the variant P_1_Q_2_P_3_Q_4_L_5_P_6_Y_7_**S_8_**Q_9_ (P to S substitution at p8) showed an anti-33-mer binding capacity and PBMC stimulation similar to that of the DQ2.5-glia-α2 canonical CD epitope. This common variant was found in polyploid and diploid species with AA genome, but was not found in BB and DD genomes. Those peptides with two mismatches, such as P_1_Q_2_P_3_Q_4_**P_5_Q_6_**Y_7_P_8_Q_9_ (substitutions L to P at p5 and P to Q at p6), **L_1_**Q_2_P_3_Q_4_L_5_P_6_Y_7_**S_8_**Q_9_ (substitutions P to L at p1 and P to S at p8), **S_1_**Q_2_P_3_Q_4_L_5_P_6_Y_7_**S_8_**Q_9_ (substitutions P to S at p1 and p8), and P_1_Q_2_P_3_Q_4_L_5_P_6_**H_7_S_8_**Q_9_ (substitutions tyrosine (Y) to H at p7, and P to S at p8) showed a CR of 30–40% with respect to the canonical CD epitope and a stimulation index (SI) from 13 to 23 for PBMC stimulation. In contrast, the replacement of P to L at p3, p6, or p8 showed no reactivity with the moAbs. Among all of these variants, P_1_Q_2_P_3_Q_4_L_5_P_6_Y_7_**S_8_**Q_9_ and P_1_Q_2_P_3_Q_4_**P_5_Q_6_**Y_7_P_8_Q_9_ were the most frequent variants of the DQ2.5-glia-α2. The modification of P to S at p8 showed high stimulation with the moAbs and PBMCs, however, two mismatches of L to P at p5 and P to Q at p6 in the same sequence caused a threefold decrease in the immunogenicity of the DQ2.5-glia-α2 canonical epitope. This change is abundant in the BBAA genome, especially in the *T. turgidum* species.

Proliferation assays for PBMC with the canonical DQ2.5-glia-α3 epitope F_1_R_2_P_3_E_4_Q_5_P_6_Y_7_P_8_Q_9_ were tested with E on p4 by tTG2-deamidation of the original Q. Several peptides released an increased stimulatory effect on T cells, such as the DQ2.5-glia-α3 variant F_1_**L**_2_P_3_Q_4_**L_5_**P_6_Y_7_P_8_Q_9_ with two mismatches. However, other variants for this epitope, with several amino acid substitutions, had no stimulatory effect on T cells, including P to S substitution at p6, Y to Q at p7, P to Q at p3, and two substitutions of arginine (R) to P at p2 and Q to H at p9 and R to P at p2 and P to A at p3 (Figure 6c). The replacement of R to L at p2 and Q to L at p5 in the variant F_1_**L**_2_P_3_Q_4_**L_5_**P_6_Y_7_P_8_Q_9_ gave it greater stimulation capacity, given that this variant was highly abundant for the DQ2.5-glia-α3 epitope, it was found in all the polyploid species and in the BB diploid genome. However, the non-abundant variant F_1_**P_2_**P_3_Q_4_**L_5_**P_6_Y_7_P_8_Q_9_ with the change of R to P at p2 and Q to L at p5 increased both the binding of the moAbs and stimulation with T cells. The variant F_1_R_2_P_3_Q_4_L_5_P_6_Y_7_**L_8_**Q_9_ with one mismatch (P to L at p8) showed T-cell stimulatory capacity and moAb binding, but was low in abundance (0.6% to 2.5%). Nevertheless, one of the most abundant variants, F_1_**S_2_**P_3_Q_4_Q_5_P_6_Y_7_P_8_Q_9_, showed no T-cell stimulatory capacity and binding of the moAb and was found in the BB diploid genome.

According to the model of HLA-DQ2, the key amino acid residues for DQ2 binding lie at positions 1, 7, and 9, with preferential residues at positions 4 and 6 [43,44]. On the other hand, Elli et al. [45] found that substitutions at positions 2, 3, 5, and 8 also profoundly affected T-cell stimulation, indicating that these residues may all interact with the T-cell receptor (TCR). Our findings showed that the change at position 2 affected T-cell stimulation in the domain DQ2.5-glia-α1, at p8 in the DQ2.5-glia-α2 domain. In addition, the changes at positions 2, 5, and 8 in the DQ2.5-glia-α3 domain profoundly affected T-cell stimulation. Our results now provide new insights into an alternative approach, since we have showed that, by introducing specific amino acid substitutions, such as Q to H, at any position, the toxicity of the three T-cell α-gliadin epitopes could be eliminated. As such, the high level of variation influencing the immunogenicity of the major CD epitopes may offer possibilities to generate new wheat lines with reduced CD-immunogenicity, which may be potentially used as starting points for the breeding of safe wheats. 

## 4. Conclusions

The results presented here about CD DQ2.5 epitopes provide the basis for the introduction and/or selection of natural amino acid substitutions to eliminate the toxicity of the α-gliadin T-cell epitopes. Our findings show that the most abundant epitope in the DQ2.5-glia-α1 domain is the CD canonical epitope. Considering the DQ2.5-glia-α2 domain, the variants P_1_Q_2_P_3_Q_4_L_5_P_6_Y_7_**S_8_**Q_9_ and P_1_Q_2_P_3_Q_4_**P_5_Q_6_**Y_7_P_8_Q_9_ are the most abundant in this domain, while F_1_**P_2_**P_3_Q_4_Q_5_P_6_Y_7_P_8_Q_9_ is the most abundant in the DQ2.5-glia-α3 domain. Moreover, the F_1_**P_2_**P_3_Q_4_Q_5_P_6_Y_7_P_8_Q_9_ variant was also the most frequent of all the sequences studied. Our data indicate that the changes of P to S and R to P may be the most representative changes and the natural introduction of Q to H at any position eliminates the toxicity of the three T-cell epitopes. These results may offer possibilities to generate wheat varieties with a reduced CD-immunogenicity. Such varieties would help to reduce the presence of immunogenic CD epitopes in wheat flour and, while not safe for consumption by patients, might help to prevent the onset of CD in people that carry genetic risk factors. Overall, the more the scientific community knows about immunogenicity of the gliadins, the closer an alternative therapy besides GFDwill be achieved.

## Figures and Tables

**Figure 1 nutrients-11-00220-f001:**
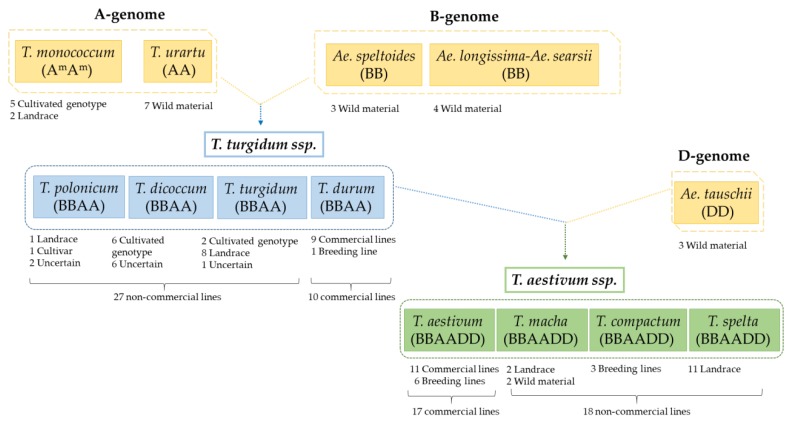
Schematic representation of 96 accessions from *Triticum* and *Aegilops* sp. showing their origin and breeding status. In total, there are thirty-five accessions of hexaploid wheats, thirty-seven accessions of tetraploid wheats, and twenty-four accessions of diploid wheats. AA, BB, and DD: diploids; BBAA: tetraploids; BBAADD: hexaploids (partially adapted from Ozuna and Barro) [3].

**Figure 2 nutrients-11-00220-f002:**
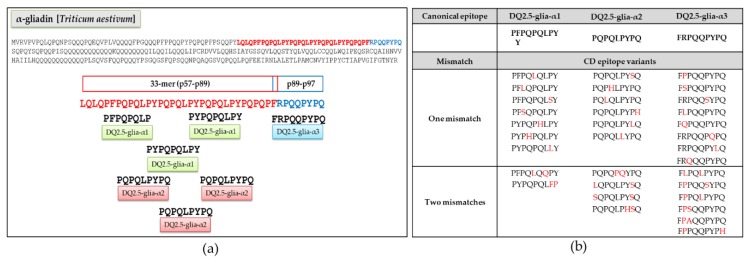
Celiac disease (CD) epitopes and variants derived from α-gliadin. (**a**) Location of canonical epitopes DQ-2.5-glia-α1, DQ-2.5-glia-α2, and DQ-2.5-glia-α3 into α-gliadin protein. (**b**) Variants of the canonical CD epitopes with one or two mismatches selected with more than 80 reads found in *Triticum* and *Aegilops* ssp. The mismatches are indicated in red.

**Figure 3 nutrients-11-00220-f003:**
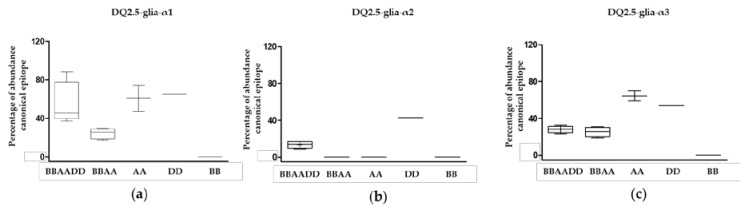
Abundance of CD canonical epitope and variants per wheat genome type. (**a**) Abundance of the canonical epitope DQ2.5-glia-α1 and variants, (**b**) abundance of the canonical DQ2.5-glia-α2 epitope and variants, and (**c**) abundance of the canonical DQ2.5-glia-α3 epitope and variants. BBAADD: hexaploid genome; BBAA: tetraploid genome; AA, DD, and BB: diploid genomes.

**Figure 4 nutrients-11-00220-f004:**
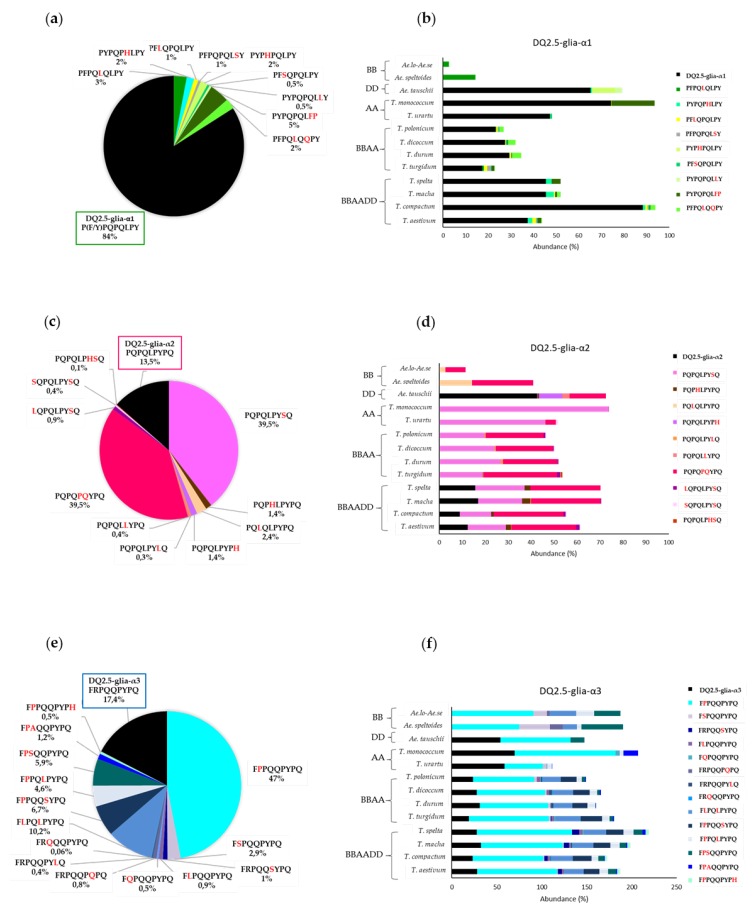
Abundance of CD canonical epitopes and variants per wheat species. (**a**) Abundance in the DQ2.5-glia-α1 epitope variants, (**b**) abundance in DQ2.5-glia-α1 epitope variants by species, (**c**) abundance in DQ2.5-glia-α2 epitope variants, (**d**) abundance in DQ2.5-glia-α2 epitope variants by species, (**e**) abundance in DQ2.5-glia-α3 epitope variants, and (**f**) abundance in DQ2.5-glia-α2 epitope variants by species.

**Figure 5 nutrients-11-00220-f005:**
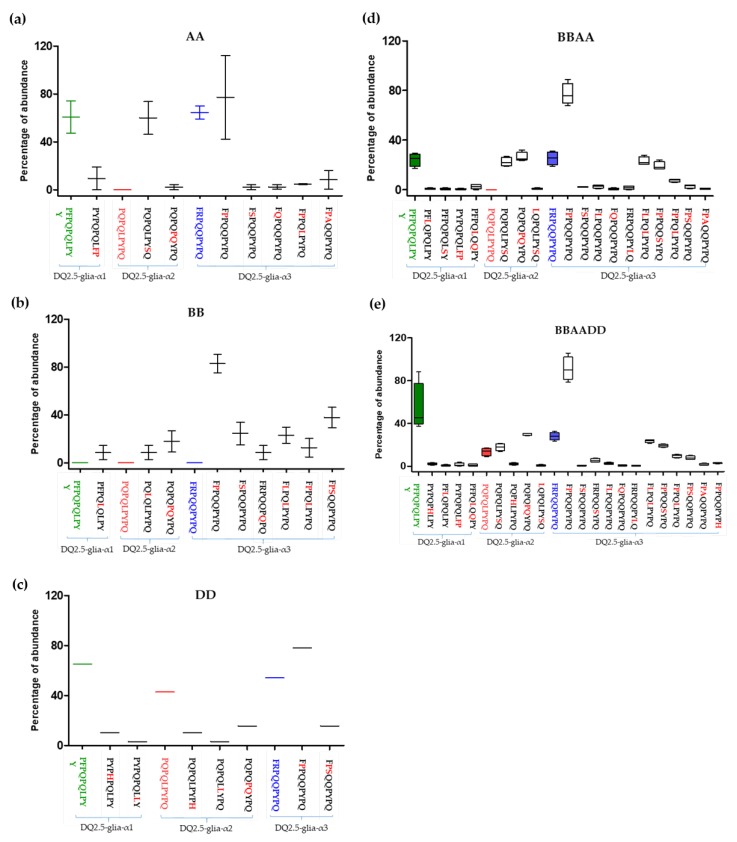
Abundance of epitope variants according to wheat genome type. The epitopes represented showed more than 1% of abundance in each gliadin domain. (**a**) AA genome, (**b**) BB genome, (**c**) DD genome, (**d**) BBAA genome, and (**e**) BBAADD genome. DQ2.5-glia-α1 epitope: green; DQ2.5-glia-α2 epitope: red; DQ2.5-glia-α3 epitope: blue.

**Figure 6 nutrients-11-00220-f006:**
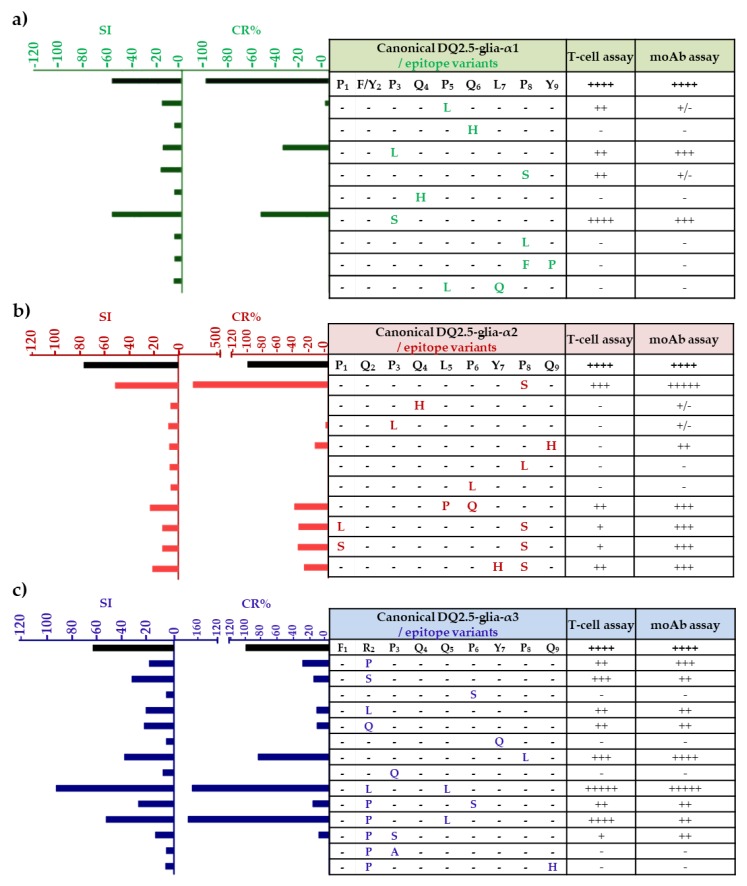
T-cell proliferation and anti-33-mer binding capacity of DQ2.5-glia-α1, DQ2.5-glia-α2, and DQ2.5-glia-α3 epitope variants and canonical epitope. (**a**) DQ-2.5-glia-α1 epitope and variants, (**b**) DQ-2.5-glia-α2 epitope and variants, and (**c**) DQ-2.5-glia-α3 epitope and variants. Variants and the canonical epitope were synthesized as deaminated 9-mer peptides to peripheral blood mononuclear cell (PBMC) assay. Proliferative responses of T cells were defined as a stimulation index (SI), which means the specific proliferation of a sample divided by the background proliferation ([PBMC + peptide]/[PBMC]). Glutamate residues (E) that would be formed by TG2-mediated deamination, which are important for recognition by T cells, are shown in italics. For the T-cell assay, the responses are represented relative to the maximum response given by the CD canonical epitope indicated by ++++. Therefore, - corresponds with 15%; + corresponds with 15–25%; ++ corresponds with 25–50%; +++ corresponds with 50–75%; ++++ corresponds with 75–100%; and +++++ corresponds with >100%. For the monoclonal antibody (moAb) assay, the amount of antigen detected is represented relative to the maximum amount (mol/L) detected in a given assay by the CD canonical epitope indicated by ++++. Therefore, - corresponds with 0%; +/- corresponds with < 5%; + corresponds with 5–10%; ++ corresponds with 10–30%; +++ corresponds to 30–60%; ++++ corresponds to 60–100%; and +++++corresponds to > 100%. CR; cross-reactivity, was calculated as follows: (IC50 of the antigen for which the moAb was raised/IC50 of each antigen assayed) × 100. The IC50 is defined as the concentration of the line that reduces the peak absorbance by 50% in the assay. Each of the letters represents the amino acid substitution of the variants.

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
