# Peer review of "Celiac Immunogenic Potential of α-Gliadin Epitope Variants from Triticum and Aegilops Species"

_nutrients, 2019, doi:10.3390/nu11020220_

Round 1
Reviewer 1 Report
The study deals with the identification of the immunogenic properties of different DQ2.5-glia-α1, DQ2.5-glia-α2, and DQ2.5-glia-α3 epitope variants per species in diploid and polyploid wheat. The study provided identification of several of the amino acid variants in the α-gliadin epitope sequences that had never been described previously, and a number never tested for their immunogenic and stimulatory capacities.
Methods and Results are clear although inevitably addressed to an expert audience.
The authors conclude that the identification of the possible natural amino-acid substitutions to eliminate the toxicity of three α-gliadin T-cell epitopes can produce wheat that maintains the technological properties of ‘regular’ wheat without the toxicity for the people with celiac disease.
Moreover, in my opinion, the data is useful for more than that reason, as the more the scientific community knows about immunogenicity of the gliadins the closer is the achievement of an alternative therapy beside gluten free diet. I suggest adding a sentence about the issue in the Discussion.
Minor
Line 109 ‘In the present study, we used the sequence data obtained in [29],’ please specify in words also what is in the ref. 29.
Fig 6. The legend should be more detailed and contain also what the acronyms stand for.
Author Response
Dear Editors,
Please find enclosed a new version of our manuscript entitled “Celiac immunogenic potential of α-gliadin epitope variants from Triticum and Aegilops species” by Ángela Ruiz-Carnicer, Isabel Comino, Verónica Segura, Carmen V. Ozuna, María de Lourdes Moreno, Miguel Ángel López-Casado, María Isabel Torres, Francisco Barro and Carolina Sousa.
We would like to thank you and the reviewers for the comments, which have helped us to improve our manuscript significantly. We have highlighted the changes using the "Track Changes" function in Microsoft Word. We agree with the revisions made by the reviewers and yourself. We have addressed all comments and criticisms made by the reviewers on a point-by-point basis. The complete modified text and definitive figures have been submitted online using the Editorial Manager system.
This manuscript has been seen and approved by all listed authors. We hope that the work is now suitable for publication in Nutrients. We thank you again for the opportunity to have a modified version of our manuscript reconsidered.
Awaiting your news,
Sincerely,
Carolina Sousa
Reviewer(s)' Comments to Author:
Reviewer: 1
We thank this reviewer for his/her precise criticisms and excellent suggestions, which have allowed us to improve our manuscript.
1) In my opinion, the data is useful for more than that reason, as the more the scientific community knows about immunogenicity of the gliadins the closer is the achievement of an alternative therapy beside gluten free diet. I suggest adding a sentence about the issue in the Discussion.
According to the reviewer´s suggestion, we have added a sentence about the issue in page 11, lines 347-349.
2) Line 109 ‘In the present study, we used the sequence data obtained in [29],’ please specify in words also what is in the ref. 29.
Following the suggestion of the reviewer, we have specified the data used in the ref. 29 (page 3, lines 109-110).
3) Fig 6. The legend should be more detailed and contain also what the acronyms stand for.
Regarding the reviewer’s recommendation, we have included in the figure legend 6 more detailed information to a better understanding (page 11, lines 321 and 330-332).
Reviewer: 2
We thank this reviewer for the positive appreciation of our manuscript.
1) More details about the anti-33-mer monoclonal antibody used should be provided-does it recognize a linear epitope or a conformational epitope? The loss of binding could be either due to the disruption of the linear epitope by the amino acid substitutions or due to the alteration of the conformation of the overall epitope?
The anti-33-mer monoclonal antibody recognizes a linear CD epitope. It was described in the references 30 and 31 (page 3, lines 113-116), therefore the loss of binding might be either due to the disruption of linear epitope by the amino acid substitutions.
Reviewer: 3
We thank this reviewer for his/her useful comment.
1) Line 18 Abstract-the sentence “The high global demand wheat and” may be better state as “The high global demand of wheat and” placing an “of” before the word “wheat”.
The reviewer’s suggestion has been incorporated into the text (page 1, lines 18).
Reviewer 2 Report
Ruiz-Carnicer et al in their manuscript titled “Celiac immunogenic potential of a-gliadin epitope variants from Triticum and Aegilops species” provide an extensive analysis of the presence and abundance of both canonical as well as variant DQ2.5-glia-a1, DQ2.5-glia-a2 and DQ2.5-glia-a3 epitopes which are the dominant drivers of immune inflammation in diploid and polyploid wheats. In addition, they have examined the topology of amino acid substitutions in the variant epitopes and have characterized the impact of these amino acid substitutions on the overall strength of immunogenicity of these variant epitopes using functional readouts including T cell proliferation and interferon gamma secretion utilizing peripheral blood collected from pediatric celiac disease subjects. The work described in this manuscript provides valuable insights to the future creation of new wheat lines with reduced immunogenicity that are safe for consumption by individuals who are genetically predisposed to develop wheat induced inflammatory responses.
Minor issue: more details about the anti-33-mer monoclonal antibody used should be provided-does it recognize a linear epitope or a conformational epitope? The loss of binding could be either due to the disruption of the linear epitope by the amino acid substitutions or due to the alteration of the conformation of the overall epitope?
Author Response

(The authors gave the same response as above.)

Reviewer 3 Report
Line 18 Abstract-the sentence “The high global demand wheat and” may be better state as “The high global demand of wheat and” placing an “of” before the word “wheat”
Author Response

(The authors gave the same response as above.)
